# Interrelation Among the Developmental Trajectories of Brain, Cognition and Behavior During Adolescence

Uwasila Binte Munir[1,3], Dawn M. Jensen[1], Aidan Troha[2], and Jingyu Liu[1,3,*]

[1]TReNDS Center, Georgia State, Georgia Tech, Emory
[2]Department of Public Health, University of Georgia, Athens, Georgia
[3]Department of Computer Science, Georgia State University, Atlanta, Georgia
[*]Corresponding author: Jingyu Liu (jliu75@gsu.edu)

*Abstract*—During adolescence, the dynamic change of brain structure and function underlies the rapid maturation of cognition and behavior. However, it remains underexplored how distinct regional brain growth patterns are related to the development curves of cognition and behavior in large, longitudinal samples. To investigate that, we employed data of 2553 children from Adolescent Brain Cognitive Developmental study, including brain morphological features, cognitive and behavioral measures from ages 9-10 to 13-14 years old. Linear mixed effects model and latent growth curve analysis were first used to estimate the development trajectories of 302 brain features, 5 cognitive and 20 behavior measures. We then applied a regularized canonical correlation analysis on the resultant trajectory characteristics of brain features and those of cognition and behavior. Two pairs of significant and robust intriguing associations were identified between brain and behavior. The first pair of components highlighted the association of the increasing rate of bilateral cerebellar volume as well as cerebral white matter volume with the decreasing rate of internalizing behavior problems. Whereas, the second pair of components emphasized the relationship between crystallized intelligence and cerebral white matter volume as well as left and right ventral diencephalon volume at baseline. These findings provide us a precise understanding of how brain maturation is associated with cognition and emotional regulation.

*Index Terms*—Brain Maturation, Adolescence, Regularized Canonical Correlation Analysis, Cognition, Behavior

## I. INTRODUCTION

Adolescence is a crucial developmental phase in which the brain undergoes extensive functional and structural reorganization [1], [2]. It is also considered pivotal for cognitive and mental health development since the basis of behavioral traits are established during this period [3], [4]. In recent years, several studies have used the Adolescent Brain Cognitive Development Consortium (ABCD) study to investigate the relationship among multiple brain features (brain morphometry, functional connectivity, and white matter integrity, etc.), cognition, and mental health, as well as environmental exposure [5]–[9]. For instance, mental health problems in adolescents were reported to be associated with smaller cortical surface area of the orbitofrontal cortex, middle temporal gyrus, and anterior cingulate cortex as well as increased family conflict [5] whereas smaller cortical volume and surface area in regions that support reading, language, spatial skills and executive functions as well as lower family income were found to be related to lower cognitive performance [6], [7]. Additionally, morphometric features as well as connectivity networks in amygdala, hippocampus and prefrontal regions were reported to be closely linked to mental illnesses [11]. However, research in this area is predominantly limited in cross-sectional settings [5]–[8], due to the challenges involved with the longitudinal data collection. Despite the fact that the dynamics of individual brain growth is closely linked to behavioral maturation [10], [11], how the developmental trajectories of cognitive and behavioral measures relate to brain maturation remains largely unexplored.

Furthermore, previous studies have focused primarily on specific brain measures or a single aspect of cognitive or behavioral problems, providing only a partial representation of the general human brain and behavioral characteristics. For instance, the coupling of two specialized brain networks (lateral frontoparietal network and default mode network) was related to the total cognitive performance [12], and distinct brain connectivities have been found to support specific cognitive functions, such as language, [13], attention [14] and episodic memory [15], etc. Separately, numerous studies have investigated the role of specialized brain networks in mental health problems [16], [17]. A comprehensive view of brain morphological development with behavior, cognition, and mental health is yet to be established, which may provide valuable insights into the regional roles of the brain in adolescent dynamic behavior and cognition, and specific risk to mental health problems.

To address this knowledge gap, we have leveraged data collected by the ABCD study [18], which has collected a wide range of longitudinal behavioral and cognitive measures and mental health assessments, as well as brain magnetic resonance imaging. We explored a comprehensive set of brain

morphological features (cortical thickness, cortical volume, surface area, sulcal depth, and subcortical volumes) derived from structural MRI. Using latent growth curve analyses of longitudinal data from 2553 adolescents, we calculated the adolescents' developmental trajectories for these brain, cognition, and behavioral variables. The resultant intercepts, slopes and/or quadratic curves (if existing) from these trajectories were included in a regularized canonical correlation analysis (rCCA) to assess their multivariate patterns of interrelations.

## II. MATERIALS AND METHODS

### A. Participants and Data

*1) Subjects:* This study analyzed data from the Adolescent Brain Cognitive Development (ABCD) cohort, a longitudinal dataset collected across 21 sites in the United States, consisting of 11,875 children aged 9–10 years recruited at baseline [18], [19]. The minimally processed dataset was utilized from Release 5.0 (https://nda.nih.gov/) of the ABCD cohort, encompassing multiple domains, including neuroimaging, demographic information, as well as cognitive and mental health assessments. The participants were filtered based on the availability and high quality of neuroimaging data across all time points available in Release 5.0., resulting in the selection of 2,553 (**1411 males and 1142 females**) participants.

*2) Neuroimaging data and features:* Imaging data (structural magnetic images, or sMRI) were collected biannually, giving us three time points for analysis; at baseline (ages 9–10) and Follow-up year 2 (ages 11-12) and Follow-up year 4 (ages 13-14) [19].. The ABCD study provides FreeSurfer (version 5.3) derived brain morphological measures, obtained from the T1-weighted sMRI images [20]. A total of 302 measures from 30 subcortical regions and 68 cortical regions from the Desikan-Killiany atlas were used, including cortical thickness, cortical volume, surface area, sulcal depth, and subcortical volumes.

*3) Cognitive and Behavioral Measures:* Adolescent cognitive performance was measured using a cognitive battery provided by the NIH Toolbox® [21]. Five tests from the NIH Toolbox were used in this study: Picture Vocabulary and Reading Recognition for assessing crystalized Intelligence (acquired knowledge/learnings), Flanker test for attention/inhibition; Pattern Comparison test for processing speed; and Picture Sequence Memory test for episodic memory, with individual scores recorded for each task. Similar to the neuroimaging measures, cognition was assessed at three time points: at baseline and at two subsequent biannual follow-ups.

Behavioral assessments were conducted annually using the Child Behavior Checklist (CBCL), which is a parent reported questionnaire for assessing a wide spectrum of behavioral and emotional problems in adolescents [22]. Empirically derived syndrome scales from eight domains were reported, including anxiety/depression, withdrawal, somatic complaints, social problems, thought problems, attention problems, rule-breaking behavior, and aggressive behavior. These scales are grouped into two higher order factors—internalizing problems (sum of anxiety, depression, withdrawing, and somatic complaints) and

externalizing problems (rule-breaking behaviour, and aggressive behaviour). Apart from the standard CBCL scales, this study also includes the DSM-5 oriented subscales (e.g., depression, ADHD, anxiety, somatic problems, oppositional defiant problems, and conduct problems) to identify the clinically relevant symptom dimensions [23]. A total of 20 behavioral problems and mental health symptom measures across five time points were included.

### B. Methods

*1) Modeling Trajectories:* In order to achieve the regional trajectories (intercept and slope) of the brain, the Linear Mixed Model (LMM) from the python package statsmodels (version 0.14.4) [24] was applied. Gender and time were included as fixed effects, morphometry measures were the dependent variables. The LMM modeled the longitudinal changes across the three time points. Distinct individual trajectories were captured using random intercepts and random slopes at the subject level to address the differences in individual brain development during adolescence. For the cognitive and behavioral features, a Latent Growth Curve Analysis (LGCA) from the lavaan package in R (version 4.4.2) [25] was used to derive the trajectories.

*2) PCA For Component Estimation:* To determine the appropriate number of latent components to use in the regularized Canonical Correlation Analysis, a principal component analysis (PCA) was applied to each modality (1: trajectories of regional brain morphometry, and 2: trajectories of cognitive-behavioral-mental health measures) separately to obtain an approximation of the true intrinsic dimensionality for each dataset. Fifteen components (explaining 85% of the cumulative variance) were retained from the cognitive and behavioral trajectories as well as 160 components (explaining 80% of the cumulative variance) from the brain, based on the elbow method.

*3) rCCA Feature Extraction and Optimization:* We used a rCCA to investigate the multivariate associations between the two modalities. CCA projects high-dimensional data into a linear space such that the correlation between the projected variables from the two datasets is maximized [26]. Given two datasets $X \in \mathbb{R}^{n \times p}$ and $Y \in \mathbb{R}^{n \times q}$, where n represents the number of subjects, p and q represents the number of variables in dataset X and Y respectively. The goal of CCA is to identify the projection vectors $u$ and $v$ that solve [27]:

$$\max_{u,v} u^{\top} \Sigma_{XY} v = \min_{u,v} \|Xu - Yv\|_2^2$$

$$\text{subject to} \quad \|Xu\|_2^2 = \|Yv\|_2^2 = 1 \tag{1}$$

Here, $\Sigma_{XY}$ denotes the covariance matrix of $XY$. L2 (Ridge) regularized CCA was employed using the python package cca_zoo [28] to reduce overfitting issues associated with CCA. The rCCA model was optimized for 15 pairs of components (lower value of component numbers embedded the two data sets) through grid search with 5-fold cross-validation over an array of regularization parameters $(\lambda_1, \lambda_2)$.

TABLE I
CANONICAL CORRELATION RESULTS FOR TRAINING, TESTING AND 100 RANDOM RESAMPLING

| | $1^{st}$ comp $(r, p)$ | $2^{nd}$ comp $(r, p)$ | $3^{rd}$ comp $(r, p)$ | $5^{th}$ comp $(r, p)$ |
|---|---|---|---|---|
| **Training** | $r = 0.57, p < 10^{-16}$ | $r = 0.46, p < 10^{-16}$ | $r = 0.38, p < 10^{-16}$ | $r = 0.36, p < 10^{-16}$ |
| **Testing** | $r = 0.51, p < 10^{-16}$ Empirical $p$-value $< 0.001$ | $r = 0.41, p < 10^{-16}$ Empirical $p$-value $< 0.001$ | $r = 0.15, p = 3 \times 10^{-4}$ Empirical $p$-value $= 0.001$ | $r = 0.14, p = 1 \times 10^{-3}$ Empirical $p$-value $= 0.008$ |
| **Freq. of significance** | 100% | 100% | 86% | 20% |

This optimized model allowed the identification of robust multivariate patterns of interrelations between the trajectories of regional brain morphometry and cognitive-behavioral functioning. The optimal values identified for $\lambda_1$ and $\lambda_2$ were 0.9 and 0.05 respectively. The objective function for the optimized rCCA model is as follows [29]:

$$\min_{u,v} \|Xu - Yv\|_2^2 + \lambda_1 \|u\|_2^2 + \lambda_2 \|v\|_2^2 \qquad (2)$$

*4) Stability Test:* To assess the generalizability of the finalized rCCA model, it was run across 100 random train-test split. Only components that were consistently significant ($P < 0.0033$) after Bonferroni correction across these splits were reported. This random sampling approach assured the robustness of the observed associations between the trajectories of brain regions and cognitive and behavioral measures across the train-test split.

## III. RESULTS

As stated earlier, a comprehensive set of 302 morphological measures from both cortical and subcortical regions were incorporated, along with five cognitive and twenty mental health measures. Trajectories for each measure were obtained using LMMs for the brain across three time points, and latent growth curve modeling for cognitive and mental health measures across three and five time points, respectively. For the linear growth models (only intercept and slope were modeled), LMM and LGCM were configured equivalently. Linear growth patterns were exhibited by brain and cognitive measures, while nonlinear growth patterns (models with intercept, slope, and quadratic terms best fit the data) were shown by behavioral measures over time.

From the derived trajectories, we have discerned that the cortical measures (volume, thickness, and surface area) tend to decrease in majority of the cortical regions except for a few (10 regions) that showed a stabilized growth pattern across time. On the contrary, the subcortical regions demonstrated upward trajectories for 15 regions (includes the cerebellar regions), slightly downward trajectories for 5 regions and others showed stable/flat trajectories. All cognitive measures showed steeply increasing slopes for both males and females. For behavioral measures, trajectories varied between males and females. For instance, males showed decreasing slopes for anxiety and somatic complaints, whereas the females exhibited U-shaped

(quadratic) trajectories for these measures. Also, externalizing problems, ADHD symptoms, social problems, oppositional defiant problems, and conduct problems were decreasing for both males and females but at different rates, while depressive and withdrawal symptoms appeared to be increasing with time. Additionally, internalizing problems and stress appeared to increase before getting stable for females, contrary to the males showing a declining slope. These nonlinear behavioral patterns may reflect underlying neural or environmental influences. For example, the decrease in anxiety and somatic symptoms in females from childhood to early adolescence [30], followed by an increase after mid-adolescence [31] may align with steady brain structural and functional maturation and then mixed with hormonal fluctuations triggered by mid-adolescence pubertal development, and/or middle school social peer pressure [32]. Similarly, the steady decline in anxiety and somatic complaints among males [33] may reflect increasing cortical maturation [34] and improved coping mechanisms shaped by socialization [35].

Three example trajectories are highlighted in Figure 1, one for each domain (95% confidence intervals are represented by the shaded regions) Figure 1(A) shows the growth trajectory of cerebral white matter volume with similar slopes for each sex, but with differences in intercepts, highlighting the expected higher volume in males. In the case of cognition, seen in Figure 1(B) as the results of the picture vocabulary test, growth shows steadily increasing slopes with no discernible variation between sexes. In Figure 1C, behavioral measure, shown here with the trajectory of somatic symptoms across time, displayed major differences in slopes between males and females despite similar intercepts, indicating diverse growth patterns with comparable initial levels for somatic symptoms.

Through the rCCA analysis, all fifteen canonical components were found to be significant ($P < 0.0033$) after Bonferroni correction in the training phase. However, when the corresponding latent variables were projected onto the test set, only four of these components ($1^{st}$, $2^{nd}$, $3^{rd}$, $5^{th}$) remained significant ($P < 0.0033$). To evaluate the stability of these canonical components, rCCA was performed across 100 random train–test splits. Among the four, the first two canonical components were consistently significant across the splits, demonstrating robust associations between brain as well as cognitive and behavioral features. Also a permutation-based

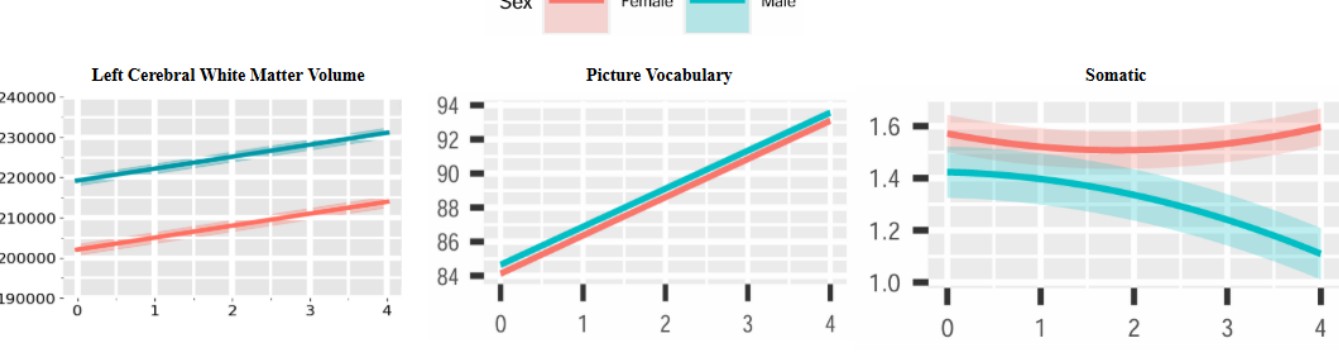

Fig. 1. Trajectories of (A) Brain regional morphological features, (B) Cognitive and (C) Behavioral measures

null distribution analysis was performed to compute empirical p-values for each component using 1000 permutations. The statistical significance of the first two components was further supported by the empirical p-values ($p < 0.001$). The canonical correlations observed in the test set for these first two components were 0.51 and 0.41 respectively, as shown in Table I. Since CCA is susceptible to overfitting even with regularization, the canonical components that remained significant in the test set across all the splits were the only ones reported hereafter.

Table II presents a comprehensive report of the brain regions and associated cognitive and behavioral measures for the first two canonical components. Based on the magnitude of the canonical loadings (a measure to determine how strongly the original variables are related to their latent components), the top three cognitive and behavioral measures and top five brain regional features are reported for each of the canonical components. The top contributing brain features are depicted in Figure 2. The first canonical component revealed an intriguing association between the dynamic brain growth of broad regions and the developmental change of multi-aspects of internalizing functioning in adolescents. Specifically, increasing slopes in the volume of the right and left cerebellum, as well as increasing slopes in the volume of bilateral cerebral white matter (absolute loadings ranging from 0.59 to 0.56 with greater stability and narrow confidence interval) were associated with decreasing quadratic curves in internalizing behavior problems that highlight somatic complaints, anxiety and depression, and overall internalizing problems.

The second canonical pair exhibited a significant connection between the baseline status (intercept) of bilateral volumes of cerebral white matter (with higher canonical loadings of 0.73 and 100% frequency), bilateral volumes of ventral diencephalon, and left orbitofrontal volume, with the baseline status of cognitive ability in the reading recognition, and picture vocabulary, as well as the linear slope of picture vocabulary. Larger volumes of associated brain regions are associated with higher scores in picture vocabulary and reading

recognition tasks.

Finally, a post hoc regression analysis was conducted including age, socioeconomic status (SES), site, sex and puberty status as covariates. While age, site, and puberty status did not show significant effects, SES and sex were significant. Importantly, the first two canonical components remained significant even after adjusting for these covariates.

## IV. DISCUSSION AND CONCLUSION

The existing studies have reported that morphological features of the orbitofrontal cortex, middle temporal gyrus, anterior cingulate cortex, amygdala, hippocampus and prefrontal regions [5]–[9] etc., are closely associated with behavioral problems and neurocognitive abilities in adolescents, based on the baseline data from the ABCD cohort. However, to the best of our knowledge, it remains unclear how the developing curvature of the behavior and cognition relates to brain maturation processes in a large longitudinal dataset. Therefore, this study explores how the trajectories of regional brain morphometry relate to behavioral and cognitive development. It not only provides insights into the regions that are sensitive to cognitive or mental health development, but also highlights how the growth in these regions might influence behavioral dynamics known in adolescence. The trajectory analyzes confirmed our hypotheses on the dramatic and consistent changes in the brain, behavior and cognition during the ages from 10 to 15 years old. While brain regions associated with gray matter volume exhibit a volumetric decline, regions associated with white matter volume show a continued increase in volume [36]. The lower cognitive regions (pericalcarine, cuneus and lingual etc) of the brain demonstrated only minimal decline or no changes in gray matter volume, indicating early maturation of the occipital lobe. In contrast, gray matter volume showed a moderate to high decline in parietal regions (e.g., superior parietal, inferior parietal, and precuneus) and frontal regions (e.g., superior frontal, lateral orbitofrontal, and medial orbitofrontal), particularly the frontal regions showing a more noticeable reduction [37]. The white matter volume of posterior regions

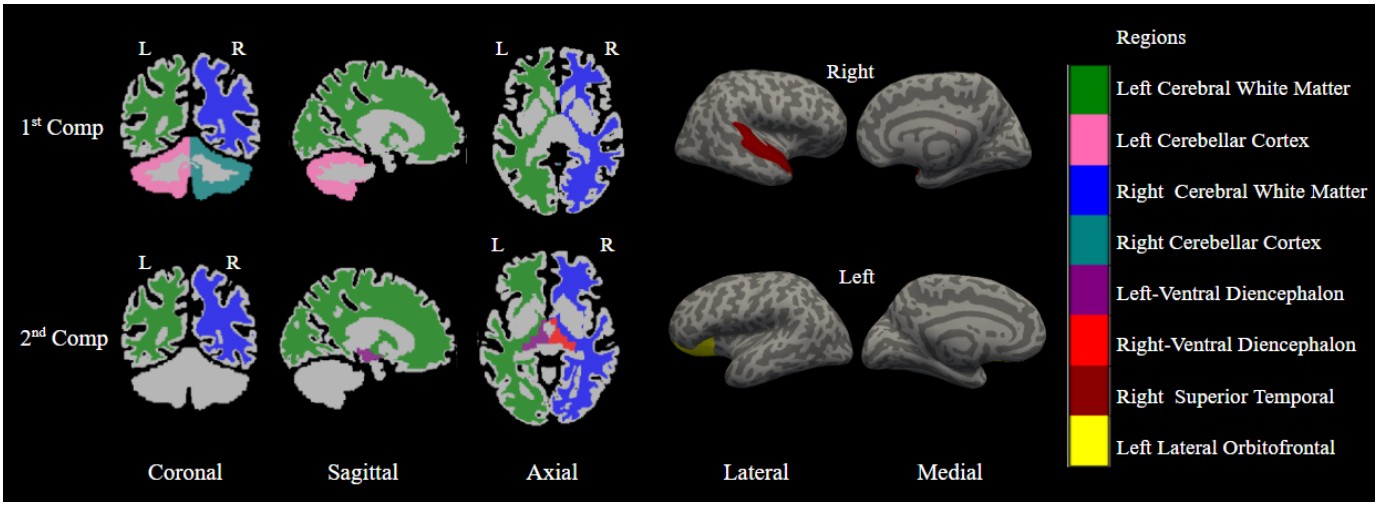

Fig. 2. Top five brain regions of first and second canonical component.

TABLE II
TOP CONTRIBUTING FEATURES AND THEIR CANONICAL LOADINGS ON THE TEST DATASET

| Canonical Pair | Canonical Variates | Features | Canonical Loadings | Stability evaluation of features across 100 iterations | |
|---|---|---|---|---|---|
| | | | | Mean Canonical Loadings (95% CI) | Frequency in Top 5 and Top 3 list |
| 1st | Brain Morphological Measures (FreeSurfer) | Left Cerebellar Cortex Volume (Slope) | -0.59 | -0.59 (-0.61, -0.56) | 99.0% |
| | | Right Cerebral White Matter Volume (Slope) | -0.58 | -0.57 (-0.65, -0.49) | 71.0% |
| | | Right Cerebellar Cortex Volume (Slope) | -0.57 | -0.56 (-0.58, -0.54) | 90.0% |
| | | Left Cerebral White Matter Volume (Slope) | -0.56 | -0.56 (-0.60, -0.51) | 63.0% |
| | | Right Superior Temporal Volume (Slope) | -0.56 | -0.55 (-0.64, -0.48) | 60.0% |
| | Cognitive-Behavioral Measures | Somatic (Quad) | 0.48 | 0.47 (0.44, 0.51) | 100.0% |
| | | Internalizing (Quad) | 0.39 | 0.38 (0.35, 0.42) | 77.0% |
| | | Anxiety/Depression (Quad) | 0.38 | 0.38 (0.35, 0.40) | 64.0% |
| 2nd | Brain Morphological Measures (FreeSurfer) | Left Cerebral White Matter Volume (Intercept) | 0.73 | 0.73 (0.68, 0.77) | 100.0% |
| | | Right Cerebral White Matter Volume (Intercept) | 0.73 | 0.73 (0.67, 0.77) | 100.0% |
| | | Left Lateral Orbitofrontal Volume (Intercept) | 0.68 | 0.68 (0.63, 0.71) | 90.0% |
| | | Left Ventral Diencephalon Volume (Intercept) | 0.68 | 0.68 (0.64, 0.71) | 100.0% |
| | | Right Ventral Diencephalon Volume (Intercept) | 0.68 | 0.67 (0.64, 0.72) | 100.0% |
| | Cognitive-Behavioral Measures | Picture Vocabulary (Intercept) | 0.78 | 0.78 (0.72, 0.83) | 100.0% |
| | | Reading Recognition (Intercept) | 0.71 | 0.71 (0.67, 0.74) | 100.0% |
| | | Picture Vocabulary (Slope) | 0.58 | 0.57 (0.52, 0.63) | 100.0% |

showed moderately increasing slopes, while anterior regions exhibited only a slight increase, suggesting late maturation of anterior white matter as well as white matter in general [38]. All cognitive measures appeared to be increasing with time as described in the prior works [39], whereas mental health measures have both increasing and decreasing trajectories, suggesting different course of development for different measures [40].

In conclusion, the rCCA analysis successfully extracted two distinct patterns of association between morphological brain maturation and cognitive-behavioral development; with the first pattern isolating the internalizing behavior and the second reflecting the cognitive aspect. In both patterns, white matter volumes were predominant, highlighting the vital role of white matter volume in the development of both behavior and cognition in adolescents [40], [41]. The first canonical pattern set highlights the effect of volumetric changes in bilateral cerebral white matter and bilateral cerebellar cortex in the emotional development of adolescents. In contrast, the second canonical set points to the sensitivity of the bilateral cerebral white matter and bilateral ventral diencephalon to crystalized cognition at baseline. In other words, individuals with greater volume in these brain regions showed better crystallized intelligence at baseline. The features of the second canonical component demonstrated higher mean absolute loadings for both brain and behavior compared to the first canonical component, indicating a greater shared variance percentage for the second canonical component. These findings give us a more pinpoint understanding of how emotional and cognitive regulation is associated with brain maturation. In this study, children were recruited at the ages of 9–10 years and followed up until the ages of 13-14 years [18], [19]. However, according to the World Health Organization, the adolescent period lasts from 10 to 19 years of age [42]. Therefore, future work should include data spanning more time points to fully capture the associations between the trajectories of brain and behavior across the entire adolescent period.

## ACKNOWLEDGMENT

This research work was funded by NIH grant: R01MH130595. Data used in the preparation of this article were obtained from the Adolescent Brain Cognitive DevelopmentSM (ABCD) Study (https://abcdstudy.org), held in the NIMH Data Archive (NDA). This is a multisite, longitudinal study designed to recruit more than 10,000 children age 9-10 and follow them over 10 years into early adulthood. Additionally, we would like to thank Aidan Troha for his previous work in analyzing cognition and behavioral trajectory.

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
