# OpenReview forum: "Interrelation Among the Developmental Trajectories of Brain, Cognition and Behavior During  Adolescence"
_IEEE.org/EMBS/BHI/2025/Conference — BHI 2025_

### Official Review · Reviewer_1AXH · 2025-07-11
**Borderline Accept – Longitudinal brain-cognition–behavior coupling in ABCD (solid dataset, but analysis and framing need tightening)**

**Confidence:** 4
**Clarity Of Writing:** good
**Clinical Significance:** fair
**Methodological Novelty:** good
**Overall Rating:** 5
**Final Rating:** 6

**Experiments And Results:**

good

**Questions For The Authors:**

1. Please add puberty status and socioeconomic status as fixed effects and report whether the first two canonical correlations still hold. A positive robustness check would raise my confidence in the results.
2. ABCD already provides Year 6 data. Do you intend to extend the trajectories and show that the same patterns persist into mid-adolescence? Demonstrating stability over a longer window would strengthen the novelty claim.
3. Please include a permutation-based null distribution to justify the extremely small p-values reported for the canonical correlations.
4. Give a concrete example of how the cerebellar and white-matter findings could guide early screening or targeted interventions, as this would enhance the clinical relevance of the work.

**Strengths:**

The study’s chief asset is its unusually large longitudinal sample, which covers more than 2,500 adolescents with three MRI waves and up to five behavioral assessments-an impressive reach for this age group. Whole-brain coverage through 302 cortical and subcortical measures permits a fine-grained anatomical view, and the authors further strengthen their analysis by combining regularized CCA with an exhaustive grid search and repeated stability checks to curb over-fitting. The work also offers a coherent developmental narrative: accelerated maturation of cerebellum and white matter appears to buffer emotional difficulties, whereas greater baseline white-matter and diencephalic volumes predict higher crystallized intelligence. Clear, reader-friendly figures-particularly the growth curves and brain heat maps-help communicate these patterns.

**Summary Of The Paper:**

The manuscript draws on data from 2,553 participants in the ABCD cohort, tracing development from ages nine or ten to thirteen or fourteen. For each child the authors model trajectories of 302 brain-morphometry variables with linear mixed models, then estimate growth patterns for five cognitive tests and twenty behavioral or mental-health scales using latent growth-curve models. They convert the resulting intercepts and slopes into inputs for a regularized canonical correlation analysis and discover two stable brain–behavior axes. The first links faster bilateral cerebellar and cerebral white-matter growth to fewer internalizing symptoms, while the second ties larger baseline white-matter and ventral diencephalon volumes to stronger crystallized-intelligence scores. These findings persist across one hundred random train-test splits, although only the first two canonical pairs remain robust.

**Weaknesses:**

Several limitations temper the contribution. The chosen age window captures only early adolescence, leaving unanswered questions about development from the mid-teens into young adulthood. Key confounding influences-pubertal status, socioeconomic background, scanner site, and racial or ethnic identity-are not included in the statistical models, which could bias the reported associations. The first two canonical correlations, around 0.51 and 0.40 on held-out data, explain only moderate variance, and subsequent components drop off sharply. Loadings are presented without permutation-based null testing or FDR correction despite the high dimensionality, making it difficult to judge statistical reliability. Clinical implications remain speculative; the manuscript speaks of “precise understanding” but stops short of outlining concrete screening or intervention strategies. Finally, typographical errors such as “exhibited” and “quadratic,” extra spaces in tables, and ambiguous caption labels detract from the paper’s polish.

---

### Official Review · Reviewer_3waM · 2025-07-11
**Overstated Claims About Brain-Behavior Associations Based on Flawed Analysis**

**Confidence:** 5
**Clarity Of Writing:** good
**Clinical Significance:** good
**Methodological Novelty:** fair
**Overall Rating:** 3

**Experiments And Results:**

fair

**Questions For The Authors:**

How do you justify the complete absence of multiple comparisons correction when testing thousands of potential associations? What would your results look like with appropriate FDR or Bonferroni correction?

Why retain 290 of 302 brain components in your PCA? This provides essentially no dimensionality reduction and likely preserves noise. Have you tested more aggressive dimensionality reduction?

Can you explain why 28 of 30 canonical components failed to replicate consistently? Doesn't this suggest fundamental overfitting issues with your approach?

Have you tested the stability of your findings with different trajectory modeling approaches or different numbers of retained components?

What happens to your results if you use more stringent significance thresholds appropriate for multiple testing?

**Strengths:**

The study leverages a substantial longitudinal dataset from the ABCD cohort with 2,553 participants tracked across multiple timepoints, providing good statistical power. The comprehensive approach examining 302 brain morphological features alongside cognitive and behavioral measures is ambitious and potentially informative. The use of regularized CCA to handle high-dimensional data and the stability testing across 100 random train-test splits shows awareness of overfitting concerns. The trajectory modeling approach using linear mixed models and latent growth curve analysis is appropriate for longitudinal data. The writing is generally clear and figures are well-presented.

**Summary Of The Paper:**

This study examines the relationship between developmental trajectories of brain morphology, cognition, and behavior in 2,553 adolescents from the ABCD study, tracked from ages 9-10 to 13-14. The authors extracted trajectories using linear mixed models for 302 brain features and latent growth curve analysis for 5 cognitive and 20 behavioral measures. They then applied regularized canonical correlation analysis (rCCA) to identify associations between brain trajectories and cognitive/behavioral trajectories. Two significant canonical components were identified: one linking cerebellar and white matter volume increases with decreasing internalizing behaviors, and another associating baseline white matter and ventral diencephalon volumes with crystallized intelligence.

**Weaknesses:**

The weakness is the multiple comparisons problem that is completely unaddressed. With 302 brain features, 25 cognitive/behavioral measures, and trajectory parameters (intercepts, slopes, quadratic terms), the authors are testing thousands of potential associations without any correction for multiple comparisons. The reported p-values are meaningless in this context.

The dimensional reduction approach is problematic. Using PCA to retain 290 components from 302 brain features (96% of original dimensions) provides virtually no dimensionality reduction and likely retains noise. The justification for retaining components to explain 95% variance is arbitrary and inappropriate for such high-dimensional data.

The rCCA methodology has flaws. The authors report 30 significant canonical components in training but only 2 remain consistently significant across test splits. This 93% failure rate strongly suggests overfitting despite regularization. The interpretation of canonical loadings as indicating causal or mechanistic relationships is unjustified - CCA identifies linear combinations that maximize correlation, not meaningful biological associations.

The trajectory modeling is inconsistent between modalities. Using different methods (LMM vs LGCA) for brain versus cognitive/behavioral measures introduces unnecessary methodological variance. The claim that these methods are "configured equivalently" for linear growth is unsupported.

---

### Official Review · Reviewer_P2HU · 2025-07-16

**Confidence:** 2
**Clarity Of Writing:** great
**Clinical Significance:** great
**Methodological Novelty:** good
**Overall Rating:** 6

**Experiments And Results:**

great

**Questions For The Authors:**

(1) Expand the discussion on nonlinear growth patterns, particularly for behavioral measures. How might these patterns reflect underlying neural or environmental factors? This could provide a more nuanced understanding of adolescent mental health development.

**Strengths:**

(1) The paper addresses a critical gap in developmental neuroscience by linking brain morphological changes with cognition and behavior in a longitudinal framework. This approach provides a more comprehensive understanding of adolescent brain maturation compared to previous cross-sectional studies.
(2) The use of data from 2,553 participants across multiple sites strengthens the generalizability of the results and ensures sufficient statistical power.

**Summary Of The Paper:**

This paper explores the dynamic interrelations between brain morphological maturation, cognitive development, and behavioral changes during adolescence using longitudinal data from the Adolescent Brain Cognitive Development (ABCD) study. The authors analyzed data from 2,553 adolescents aged 9–10 to 13–14 years, incorporating brain imaging, cognitive assessments, and behavioral measures. By modeling developmental trajectories of brain features (302 measures), cognition (5 tests), and behavior (20 measures), the study applied regularized canonical correlation analysis (rCCA) to identify multivariate associations between these domains.

**Weaknesses:**

(1)  The paper acknowledges nonlinear trajectories for behavioral measures but does not explore their implications in depth. For instance, U-shaped trajectories for anxiety and somatic complaints in females suggest complex developmental processes that warrant further interpretation.

---

### Official Review · Reviewer_Bwua · 2025-07-17
**Interrelation Among the Developmental Trajectories of Brain, Cognition and Behavior During Adolescence**

**Confidence:** 3
**Clarity Of Writing:** good
**Clinical Significance:** good
**Methodological Novelty:** good
**Overall Rating:** 7

**Experiments And Results:**

good

**Questions For The Authors:**

Given the insights gained from your work on brain development trajectories, do you foresee the possibility of developing a predictive model for the neurological progression in children with neurodegenerative diseases over time? Furthermore, have you considered the potential for extrapolating these findings to predict brain development and decline at older ages?

**Strengths:**

The work is well-structured and clearly written, providing a good description of the methodologies employed in its development.

**Summary Of The Paper:**

This study investigates the interrelation among the developmental trajectories of brain structure, cognition, and behavior during adolescence. The researchers used linear mixed-effects models and latent growth curve analysis to estimate the developmental trajectories of 302 brain morphological features, 5 cognitive measures, and 20 behavioral measures. A regularized canonical correlation analysis was then applied to these trajectory characteristics to identify significant associations. The findings revealed two robust associations: first, an increasing rate of bilateral cerebellar and cerebral white matter volume was linked to a decreasing rate of internalizing behavior problems and second, crystallized intelligence showed a relationship with cerebral white matter volume and the volume of the left and right ventral diencephalon at baseline.

**Weaknesses:**

I would appreciate a more detailed discussion of the biological implications of the obtained results, as well as a state-of-the-art discussion.

---

### Official Review · Reviewer_MHGN · 2025-07-21
**Developmental trajectory of brain for behavioral and cognitive development**

**Confidence:** 5
**Clarity Of Writing:** great
**Clinical Significance:** fair
**Methodological Novelty:** fair
**Overall Rating:** 5

**Experiments And Results:**

good

**Questions For The Authors:**

Its not clear where the novelty of this paper lies, elaborate on what is the novelty of this research vs the literature using the same dataset with morphometric and behavioral data
What is the significance of this research, expand on this in the discussion/conclusion section.

**Strengths:**

Its an important area of research.

**Summary Of The Paper:**

The paper explores the trajectory of brain morphometry involved in behavioral and cognitive development in adolescents using two timepoints. The paper also looks at their cognitive and behavioral measures to investigate the relationship between the change in morphometry of brain region and cognitive/behavioral measures.

**Weaknesses:**

-The study need a deep literature review of the topic and studies published with the same dataset that used brain morphometry. They need to highlight their novelty and what other studies have done and what is the significance of their findings